# AD-VAT: An asymmetric dueling mechanism for learning visual active tracking

**Fangwei Zhong♣, Peng Sun♦, Wenhan Luo♦, Tingyun Yan♣, Yizhou Wang♣♠♡◇**

♣ Nat'l Eng. Lab. for Video Technology, Key Lab. of Machine Perception (MoE),
  Computer Science Dept., Peking University
♦ Tencent AI Lab
♠ Cooperative Medianet Innovation Center
♡ Peng Cheng Lab
◇ Deepwise AI Lab
{zfw, yanty18, yizhou.wang}@pku.edu.cn,
{pengsun000, whluo.china}@gmail.com

## Abstract

Visual Active Tracking (VAT) aims at following a target object by autonomously controlling the motion system of a tracker given visual observations. Previous work has shown that the tracker can be trained in a simulator via reinforcement learning and deployed in real-world scenarios. However, during training, such method requires manually specifying the moving path of the target object to be tracked, which cannot ensure the tracker's generalization on unseen object moving patterns. To learn a robust tracker for VAT, in this paper, we propose a novel adversarial RL method which adopts an Asymmetric Dueling mechanism, referred to as AD-VAT. In AD-VAT, both the tracker and the target are approximated by end-to-end neural networks, and are trained via RL in a dueling/competitive manner: *i.e.*, the tracker intends to lockup the target, while the target tries to escape from the tracker. They are asymmetric in that the target is aware of the tracker, but not vice versa. Specifically, besides its own observation, the target is fed with the tracker's observation and action, and learns to predict the tracker's reward as an auxiliary task. We show that such an asymmetric dueling mechanism produces a stronger target, which in turn induces a more robust tracker. To stabilize the training, we also propose a novel partial zero-sum reward for the tracker/target. The experimental results, in both 2D and 3D environments, demonstrate that the proposed method leads to a faster convergence in training and yields more robust tracking behaviors in different testing scenarios. For supplementary videos, see: https://www.youtube.com/playlist?list=PL9rZj4Mea7wOZkdajK1TsprRg8iUf51BS

## 1 Introduction

Visual Active Tracking (VAT) aims at following a target object by autonomously controlling the motion system of a tracker given visual observations. VAT is demanded in many real-world applications such as autonomous vehicle fleet (*e.g.*, a slave-vehicle should follow a master-vehicle ahead), service robots and drones (*e.g.*, a drone is required to follow a person when recording a video). To accomplish the VAT task, one typically needs to perform a sequence of tasks such as recognition, localization, motion prediction, and camera control. However, conventional visual tracking (Babenko et al., 2009; Ross et al., 2008; Mei & Ling, 2009; Hu et al., 2012; Bolme et al., 2010; Kalal et al., 2012) aims to solely propose a 2D bounding box of the target frame by frame, and does not actively take into consideration the control of camera. Thus, compared to the problem of "passive" tracking, VAT is more practical and challenging.

With the advancement of deep reinforcement learning (Sutton & Barto, 1998; Mnih et al., 2015; Silver et al., 2016; Mnih et al., 2016), training an end-to-end deep neural network via reinforcement learning for VAT is shown to be feasible (Luo et al., 2018; Luo et al., 2019). The authors learn a policy that maps raw-pixel observation to control signal straightly with a Conv-LSTM network. Such an end-to-end approach could save the effort of tuning an extra camera controller. Meanwhile, it also outperforms the conventional methods where the passive tracker is equipped with a hand-engineered camera controller.

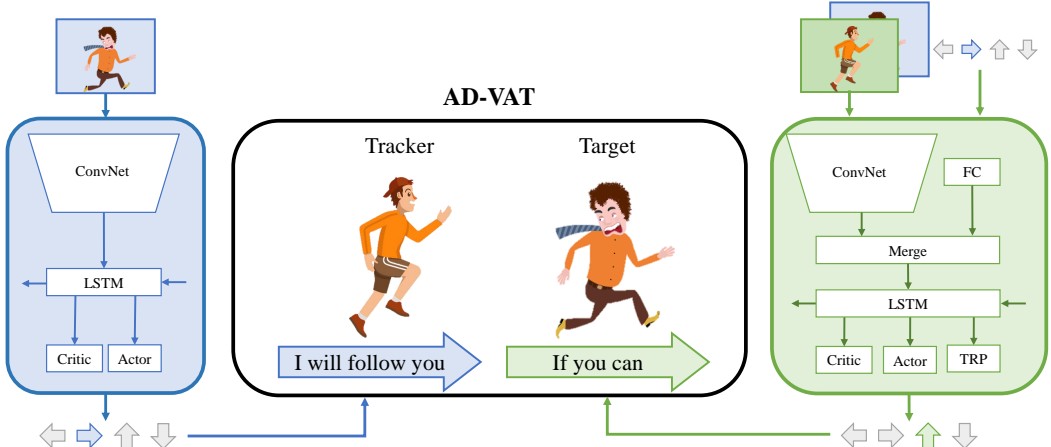

Figure 1: An overview of AD-VAT. Note that TRP is *Tracker Reward Predictor*.

However, the performance of the deep reinforcement learning based tracker is still limited by the training methods. Due to the "trial-and-error" nature of reinforcement learning, it is infeasible to directly train the tracker in the real world. Alternatively, virtual environments are always utilized to generate sufficient data for training without tedious human labeling. Nevertheless, to deploy the trained tracker in the real world, one has to overcome the virtual-to-real gap. One solution can be building numbers of high-fidelity environments (Zhu et al., 2017). However, it is expensive and tedious to build such environments for VAT. Both the visual rendering (illumination, texture, *etc.*) and the physical properties should be carefully designed to emulate the real world. Suppose we carry out VAT where the target is a pedestrian. To build the environment, one has to not only model the human's appearance, but also design physical rules and the pedestrian's trajectory so that it moves naturally like a human beings. Recently, (Luo et al., 2018) tried to overcome the virtual-to-real gap by applying the so-called environment augmentation technique. They diversify the visual appearance by changing the placement of the background objects and by flipping left-right the screen frame. However, they neglect another important factor, that is, *the motion of the target* for VAT task. Intuitively, the complexity and diversity of the target motion in training will impact the generalization of the data-driven tracker. For example, if the target only moves forward during training, the tracker may over fit to move straightly and fail to track other motion patterns, like a sharp turn.

In this work, we propose a novel adversarial RL method for learning VAT, refereed to as AD-VAT (*Asymmetric Dueling mechanism for learning Visual Active Tracking*). In the mechanism, the tracker and the target object, viewed as two learnable agents, are opponents and can mutually enhance during competition (See Fig. 1). As the training proceeds, the environments of AD-VAT naturally compose a curriculum, because the tracker is more likely to compete with a target with the appropriate difficulty level when both agents are becoming stronger simultaneously. When exploring the escape policy, the target consequently generates various trajectories to train the tracker. By the dueling/adversarial mechanism, the target is encouraged to discover the weakness of the tracker more often, which could serve as a kind of "weakness-finding" that makes the tracker more robust. However, in practice, using heuristic adversarial RL method for training VAT is unstable and slow to converge.

To address these issues, we derive two components in AD-VAT: *partial zero-sum reward(PZR)* and *tracker-aware model(TAM)* for target. PZR is a hybrid reward structure. It encourages a zero-sum tracker-target competition in the near range, where the target is close to the expected position to tracker; whereas, beyond the zero-sum zone, it is a non-zero-sum game, in which the target is penalized for running too far to track. Such reward structure is inspired by an observed phenomenon, that when the target quickly learns to be far away from the tracker while the tracker has no chance to see the target once more and henceforth gets plateaus during training. To learn the optimal policy to escape, we model the target with a "tracker-aware network", *i.e.*, besides its own observation, the observation and actions of the tracker are also fed to the escaping network. In addition, to shape a better representation about the tracker, we add an auxiliary task for the target, learning to predict the immediate reward of the tracker. We argue that such an "asymmetric dueling" mechanism is able to learn a stronger target, which vice versa yields a more robust tracker ultimately.

The experiment is conducted in various 2D and 3D environments for further studying AD-VAT. The 2D environment is a matrix map where obstacles are randomly placed. In the 2D environments, we evaluate and quantify the effectiveness of our approach in an ideal condition, free from noise in observation and action. We also conduct an ablation study to show the effectiveness of the two important components, "partial zero-sum reward" and "tracker-aware network". The 3D environments are built on Unreal Engine 4, a popular game engine for building high-fidelity environments. We choose a large room for training, where the texture of the background/players and the illumination are randomized. Three realistic scenarios built by artists are used for further evaluating the robustness. In the 3D environments, we further demonstrate that the tracker trained in AD-VAT is capable of generalizing to high-fidelity environments even it is trained in a simple environment.

The contributions of our work can be summarized as follows:

- We propose a novel Adversarial Reinforcement Learning method for VAT task, *i.e.*, the Asymmetric Dueling mechanism (AD-VAT). In AD-VAT, the target learns to generate diverse trajectories when competing with the tracker, which in turn helps train a more robust tracker.

- We provide two techniques to guarantee an efficient yet effective AD-VAT. 1) A partial zero-sum reward structure, which significantly stabilizes the training. 2) A tracker-aware network for the target, which yields better escaping policy and consequently better tracking policy.

## 2 RELATED WORK

**Active Object Tracking.** As described above that, active object tracking deals with object tracking and camera control at the same time. This problem attracts less attention compared with traditional object tracking (or visual object tracking) (Wu et al., 2013). In general, this problem could be addressed in a two-step manner or in an end-to-end manner. In the two-step solution, traditional object tracking and camera control are conducted sequentially to obtain tracking results and manipulate camera. Great progress has been achieved in traditional object tracking in recent decades (Babenko et al., 2009; Ross et al., 2008; Mei & Ling, 2009; Hu et al., 2012; Bolme et al., 2010; Kalal et al., 2012). Thus one can utilize mature visual tracking algorithms (Henriques et al., 2015; Ma et al., 2015; Choi et al., 2017) to accomplish the passive tracking task. According to the tracking results, camera control module could be developed to actively follow a target. For example, in (Denzler & Paulus, 1994) a two-stage method is proposed to handle robot control by motion detection and motion tracking. Kim *et al.*(Kim et al., 2005) detect moving objects and track them using an active camera with pan/tilt/zoom. In (Hong et al., 2018), two modules, a perception module and a control policy module, are learned separately to train an agent accomplishing both an obstacle avoidance task and a target following task.

Admittedly, tracking algorithms have been successful while still not perfect. Camera control based on tracking results is challenging due to factors such as that the correspondence between image space and camera parameter space is unknown. Additionally, joint tuning of visual tracking and camera control is expensive and encounters trial-and-errors in real world. In end-to-end methods, direct mapping between raw input frame and camera action is established. Thus the intermediate visual tracking results are not necessarily required. For instance, by reinforcement learning, camera is controlled by signal outputted from a Conv-LSTM network given raw input frame (Luo et al., 2018; Luo et al., 2019). This end-to-end solution verifies the effectiveness, while not efficient enough to solve this problem. To improve the generalization potential, environment augmentation is conducted in this work. However, the target itself, like the path, motion pattern is fixed in their augmented environments, which is believed to limit the performance. This inspires us to resort to the idea of dueling in this paper, *i.e.*, the target learns to get rid of the tracker by itself, guiding the learning of the tracker.

**Adversarial Reinforcement Learning.** Using an adversarial framework to improve the RL agent's robustness is not a new concept. In (Huang et al., 2017), they add nonrandom adversarial noise to state input for the purpose of altering or misdirecting policies. Mandlekar *et al.* (Mandlekar et al., 2017) use adversarial examples to actively choose perturbations during training in physical domains. In (Pinto et al., 2017), an adversary is introduced to apply adversarial force during training. It produces an agent whose policy is robust to a wide range of noise at testing time. Sukhbaatar *et al.* (Sukhbaatar et al., 2017) pit a virtual agent Alice against another agent Bob, where Alice

creates steadily more difficult challenges for Bob to complete. In (Held et al., 2017), they optimize a generator network via adversarial training to make an agent automatically produce tasks that are always at the appropriate level of difficulty for the agent. Roughly speaking, in these proposed methods the adversary is viewed as a ghost/virtual player, which is unseen and could only challenge the protagonist by adding noise in the observation (Huang et al., 2017), action (Pinto et al., 2017) and the system dynamics (Mandlekar et al., 2017), or by generating the goal/initial position for navigation task (Held et al., 2017; Sukhbaatar et al., 2017). In this paper, we design a two-agent no-cooperative game for VAT task, where the tracker intends to lockup the target, while the target tries to escape from the tracker. Unlike the aforementioned previous work, the adversary (target to be tracked) in AD-VAT is a physical player, which could fully control the movement of the target at any time step. In this paper, such two-physical-player-competition is referred to as "dueling". We argue that such a fully controllable opponent could bring to the protagonist more challenges during training and henceforth produces a more robust visual tracker.

The proposed approach is also related to self-play (Silver et al., 2016; 2017; Bansal et al., 2017). In self-play, two physical players usually compete for the same goal, with symmetric observation and action space. Usually, both players share the same model so as to ensure each agent seeing an environmental dynamics at appropriate difficulty level. It is thus viewed as a means to automatically generate learning curriculum (Bengio et al., 2009; Kumar et al., 2010). Our setting is substantively different from standard self-play. In AD-VAT, the two players are asymmetric in observation and task. The target observes more, and is equipped with additional auxiliary task for its adversarial policy. Thus, the target and tracker could not share the same model while learning in AD-VAT, which would make the learning become unstable. To address the issue, we propose two techniques which stabilize the training, as described in Sec. 3.

## 3 METHOD

In this section, we introduce our proposed method: *Asymmetric Dueling mechanism for learning Visual Active Tracking (AD-VAT)*. At first the proposed method is formulated as a two-player game. Then we illustrate the two key components in AD-VAT: partial zero-sum reward structure and a tracker-aware model for the target.

### 3.1 FORMULATION

We adopt the Partial Observable Two-Agent Game settings (Srinivasan et al., 2018), which extends the Markov Game (Littman, 1994) to partial observation. For the notations of our two-agent game, let subscript 1 denote the tracker (agent 1) and subscript 2 denote the target (agent 2). The game is governed by the tuple $< \mathcal{S}, \mathcal{O}_1, \mathcal{O}_2, \mathcal{A}_1, \mathcal{A}_2, r_1, r_2, \mathcal{P} >$, where $\mathcal{S}, \mathcal{O}, \mathcal{A}, r, \mathcal{P}$ denote state space, observation space, action space, reward function and environment state transition probability, respectively. Let subscript $t \in \{1, 2, ...\}$ denote the time step. In the case of partial observation, we have the observation $o_{1,t} = o_{1,t}(s_t, s_{t-1}, o_{t-1})$, where $o_t, o_{t-1} \in \mathcal{O}$, $s_t, s_{t-1} \in \mathcal{S}$. It reduces to $o_{1,t} = s_t$ in case of full observation. The counterpart notation $o_{2,t}$ is defined likewise. When the two agents take simultaneous actions $a_{1,t} \in \mathcal{A}_1, a_{2,t} \in \mathcal{A}_2$, the updated state $s_{t+1}$ is drawn from the environment state transition probability $\mathcal{P}(\cdot|s_t, a_{1,t}, a_{2,t})$. Meanwhile, the two agents receive rewards $r_{1,t} = r_{1,t}(s_t, a_{1,t}), r_{2,t} = r_{2,t}(s_t, a_{2,t})$. The policy of the tracker, $\pi_1(a_{1,t}|o_{1,t})$, is a distribution over tracker action $a_{1,t}$ conditioned on its observation $o_{1,t}$. We rely on model-free independent Reinforcement Learning to learn $\pi_1$. Specifically, the policy takes as function approximator a Neural Network with parameter $\theta_1$, written as

$$\pi_1(a_{1,t}|o_{1,t}; \theta_1). \tag{1}$$

Likewise, the policy of the target can be written as

$$\pi_2(a_{2,t}|o_{2,t}; \theta_2). \tag{2}$$

Note that we further extend the policy to a tracker-aware policy as Eq.(7). The tracker intends to maximize its expected return

$$\mathbb{E}_{\pi_1, \pi_2}\left[\sum_{t=1}^{T} r_{1,t}\right] \tag{3}$$

by learning the parameter $\theta_1$, where $T$ denotes the horizon length of an episode and $r_{1,t}$ is the immediate reward of the tracker at time step $t$. In contrast, the target tries to maximize

$$\mathbb{E}_{\pi_1,\pi_2}\left[\sum_{t=1}^{T} r_{2,t}\right] \tag{4}$$

by learning $\theta_2$.

## 3.2 REWARD STRUCTURE

The conventional adversarial methods (Bansal et al., 2017; Pinto et al., 2017) usually formulate the policy learning as a zero-sum game. In the zero-sum game, the sum of the reward of each agent is always 0, *e.g.*, $r_{1,t} + r_{2,t} = 0$. However, such kind of formulation is not suitable for VAT. Considering a case that, when the two opponents are too far to observe each other, their taken actions can hardly influence the observation of their opponents directly under the partial observable game. In this case, the sampled experiences are usually meaningless and ineffective for improving the skill level of the agent. So constraining the competition in the observable range would make the learning more efficient. Motivated by this, we shape a partial zero-sum reward structure, which utilizes the zero-sum reward only when the target is observed by the tracker, but gives penalties to each agent when they are far. In the following, we will introduce the details of the partial zero-sum reward structure for visual active tracking.

**Reward for tracker.** The reward for tracker is similar to that in (Luo et al., 2018), composing of a positive constant and an error penalty term. Differently, we do not take the orientation discrepancy between the target and the tracker into consideration. Considering the model of the camera observation, we measure the relative position error based on a polar coordinate system, where the tracker is at the origin $(0, 0)$. In this tracker-centric coordinate system, the target's real and expected position are represented by $(\rho_2, \theta_2)$ and $(\rho_2^*, \theta_2^*)$, respectively. Note that $\rho$ is the distance to the tracker, $\theta$ is the relative angle to the front of the tracker. With a slight abuse of notation, we can now write the reward function as

$$r_1 = A - \zeta\frac{|\rho_2 - \rho_2^*|}{\rho_{max}} - \xi\frac{|\theta_2 - \theta_2^*|}{\theta_{max}}, \tag{5}$$

here $A > 0$, $\zeta > 0$, $\xi \geq 0$ are tuning parameters, $\xi = 0$ in the 2D environment. We do not use the direction error as part of the penalty, in the reason that the observation is omnidirectional in the 2D environments. $\rho_{max}$ is the max observable distance to the tracker. $\theta_{max}$ is the max view angle of the camera model. which equals to the Field of View (FoV). Besides, the reward is clipped to be in the range of $[-A, A]$ to avoid over punishment when the object is far away from the expected position.

**Reward for target.** The reward for the target object is closely related to the reward of the tracker, written as:

$$r_2 = -r_1 - \mu \cdot \max(\rho_2 - \rho_{max}, 0) - \nu \cdot \max(|\theta_2| - \frac{\theta_{max}}{2}, 0), \tag{6}$$

where $r_1$ is the reward of the tracker as defined in Eq. (5), $\mu > 0$, $\nu \geq 0$ are tuning parameters controlling the factor of each penalty term. $\nu$ is 0 in the 2D environment, as the angular penalty factor $\xi$ in Eq. (5). The target is in the nearly observable range, where $\rho_2 < \rho_{max}$ and $|\theta_2| < \theta_{max}$. In the observable range, the reward function is simplified to $r_2 = -r_1$, which means that the target and tracker play a zero-sum game. When the target gets out of the observable range, the penalty term will take effect on the reward. The farther the target goes out of the range, the larger the penalty it gets. By applying this reward function, the optimal policy for the target we expect should be escaping and disappearing from the observable range of the tracker but keeping close to the edge of the range. $r_2$ is also clipped in the range of $[-A, A]$. Furthermore, we provide the details of each parameters and the visualization of the $r_1 + r_2$ in Appendix A .

## 3.3 TRACKER-AWARE TARGET

By tracker-awareness, the target would be "stronger" than the tracker, as it knows what the tracker knows. This idea manifests an ancient Chinese proverb, "know the enemy, know yourself, and in every battle you will be victorious", from the masterpiece Sun Tzu on the Art of War Sun (2008). The conventional adversary usually uses only its own observation Bansal et al. (2017) or shares the same observation as the protagonist Pinto et al. (2017). Recall the target policy $\pi_2(a_{2,t}|o_{2,t}; \theta_2)$ written as in Eq. (2). However, the imperfect/partial $o_{2,t}$ observation seriously degrades the performance of the adversary. Thus, we propose a "tracker-aware" model for the target. Besides the target's own observation, we additionally feed the observation and action from the tracker into the target network, in order to enrich the input information of the target. Moreover, we add an auxiliary task, which

predicts the immediate reward of the tracker (see the TRP module in Fig. 1). This auxiliary task can be treated as a kind of "opponent modeling", and alleviate the difficulty in its own policy learning. By doing so, we can write the output heads of such a "tracker-aware" policy as:

$$\pi_2\left(a_{2,t}, \hat{r}_{1,t} | o_{1,t}, a_{1,t}, o_{2,t}; \theta_2\right) \tag{7}$$

where $\hat{r}_{1,t}$ is the predicted immediate reward for the tracker and $o_{1,t}$, $a_{1,t}$ are respectively the observation and the action of the tracker. Empirical results show that the tracker-aware target yields more diversified escaping policy and finally helps producing a more robust tracker. Note that we cannot apply the trick for a tracker, as the tracker has to use its own observation during testing/deployment.

## 4 EXPERIMENTS

The following experiments explore our approach for the VAT task from 2D to 3D environments.

### 4.1 ENVIRONMENTS

**2D Environments.** Although 2D environments exhibit unreality to some extent, they are ideal for evaluating and quantifying the effectiveness of each method, sidestepping the uncontrolled noise in observation and action. In the 2D Environment, maps are represented by a $80 \times 80$ matrix, where $0$ denotes free space, $1$ denotes an obstacle, $2$ denotes the tracker, and $4$ denotes the target. We randomly generate the maps of two patterns, "maze" and "block" (see examples in the top row of Fig. 2). We use the "block" maps for training and both kinds of maps for testing. The observation of each agent is a matrix of size $13 \times 13$ around the agent. The tracker's goal is placing the target as close to the center of the observed matrix as possible. During each episode, the tracker starts from a free space in the map randomly, and the target starts around the tracker in a $3 \times 3$ tracker-centric matrix. At each step, the agent could take an action to move toward one of four directions. The experiments in the 2D environments are dedicated to evaluate and quantify the effectiveness of our approach in ideal conditions.

**3D Environments.** The 3D environments show high fidelity, aiming to mimic the real-world active tracking scenarios. The 3D environments are built on the Unreal Engine, which and could flexibly simulate a photo-realistic world. We employ UnrealCV (Qiu et al., 2017), which provides convenient APIs, along with a wrapper (Zhong et al., 2017) compatible with OpenAI Gym (Brockman et al., 2016), for interactions between RL algorithms and the environment. The observation is an image of the first-person view of the world as seen by the agent. The actions space is discrete with seven candidate motions, *move-forward, move-backward, turn-left, turn-right, turn-left-and-move-forward, turn-right-and-move-forward*, and *no-op*. For training, we build a *Domain Randomized Room(DR Room)* with two controllable players (target and tracker). The domain randomization techniques could help agents learn better feature representation in terms of visual observation. In testing, we focus on the transferring ability of the tracker to different unseen environments. We use three realistic scenarios, *Urban City, Snow Village* and *Parking Lot*, to mimic real-world scenes for evaluating. The bottom row in Fig. 2 shows the snapshots of the four 3D environments used.

Details of the four environments are:

- *DR Room* is a plain room comprised of floor and walls only, but the textures and illumination conditions are randomized. For the textures, we randomly choose pictures from a texture dataset (Kylberg, 2011) and place them on the surface of the walls, floor, and players. For the illumination condition, we randomize the intensity and color of each light source as well as each position, orientation.

- *Urban City* is a high-fidelity street view of an urban city, including well-modeled buildings, streets, trees and transportation facilities. Besides, there are some puddles on the road, reflecting the objects and buildings.

- *Snow Village* consists of bumpy snowfields with several trees, bushes and some cabins. Occasionally, the target will be occluded by trees and bushes, and the tracker will be distracted by the snowflake and halo.

- *Parking Lot* is an underground parking lot with complex illumination condition. The lack of light source makes the illumination uneven, *i.e.*, some places are bright but the others are dark. Besides, the pillars may occlude the target to track.

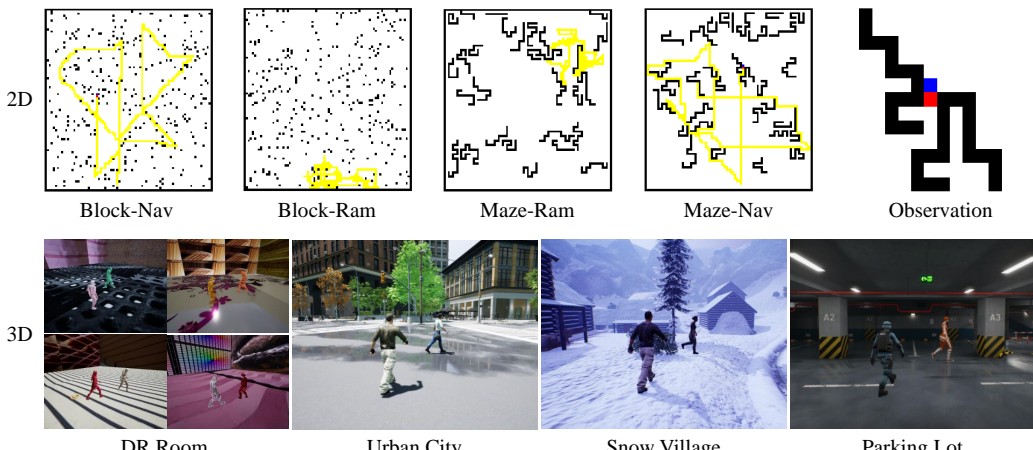

| 2D | Block-Nav | Block-Ram | Maze-Ram | Maze-Nav | Observation |

| 3D | DR Room | Urban City | Snow Village | Parking Lot |

Figure 2: The environments used for training and testing.

## 4.2 BASELINES

We provide two kinds of base target agent to randomly generate trajectories as baselines to compare with, *Rambler (Ram)* and *Navigator (Nav)*. Agent *Ram* walks randomly without any purpose like a man. Technically, it randomly samples actions from the action space and keeps executing the action $n$ times, where $n$ is also a random integer in the range of $(1, 10)$. Agent *Nav* is a navigator, which plans the shortest path to a specific goal. Thus, it could navigate to most of free space in the map. To randomize the trajectories, the goal coordinate and the initial coordinate are randomly sampled. In most of case, *Ram* prefers to walk around a local area repeatedly. In contrast, *Nav* would like to explore the map globally, shown as the yellow trajectories in Fig. 2. Thus we regard trajectories from the *Ram* as easier cases, and trajectories from *Nav* as more difficult cases for tracker.

## 4.3 IMPLEMENTATION DETAILS

Each agent is trained by A3C (Mnih et al., 2016), a commonly used reinforcement learning algorithm. The code for A3C is based on a pytorch implementation (Griffis). Multiple workers are running in parallel when training. Specifically, 16 workers are used in the 2D experiment, and 4 workers are used in the 3D experiment.

**Network Architecture.** For the tracker, we follow the end-to-end Conv-LSTM network architecture as (Luo et al., 2018). Differently, there is no fully-connected layer between the Conv-Net and the LSTM-Net in this paper. The Conv-Net is a two-layer CNN for the 2D experiments and four-layer CNN for the 3D experiments. In the 3D experiments, the input color images are transformed to gray image and the pixel values are scaled to $[-1, 1]$. we also develop the same Conv-LSTM network architecture for the target, but different in the input and output, shown as Fig. 1. The network parameters are updated with a shared Adam optimizer.

**Hyper Parameters.** For the tracker, the learning rates $\delta_1$ and $\delta'_1$ in 2D and 3D environments are 0.001 and 0.0001, respectively. The reward discount factor $\gamma = 0.9$, generalized advantage estimate parameter $\tau = 1.00$, and regularizer factor for tracker $\lambda_1 = 0.01$. The parameter updating frequency $n$ is 20, and the maximum global iteration for training is 150K. Comparing to the tracker, a higher regularizer factor is used for encouraging the target to explore, $\lambda_2 = 0.2$ in 2D and $\lambda'_2 = 0.05$ in 3D. The more exploration taken by target, the more diverse the generated trajectories are. It is useful for the learning of the tracker. Validation is performed in parallel and the best validation network model is applied to report performance in testing environments. Note that the validation environment is of the same settings as training, except that the target is controlled by a *Nav* agent. Compared with the *Ram* agent, the *Nav* agent is more challenging, thus is more suitable for validation.

**Metric.** Two metrics are employed for the experiments. Specifically, *Accumulated Reward* (AR) and *Episode Length* (EL) of each episode are calculated for quantitative evaluation. *AR* is a comprehensive metric, representing the tracker's capability about precision and robustness. It is effected by the immediate reward and the episode length. Immediate reward measures the goodness of tracking, and EL roughly measures the duration of good tracking. Because the episode is terminated when the tracker loses the target for continuous 10 steps or reaches the max episode length.

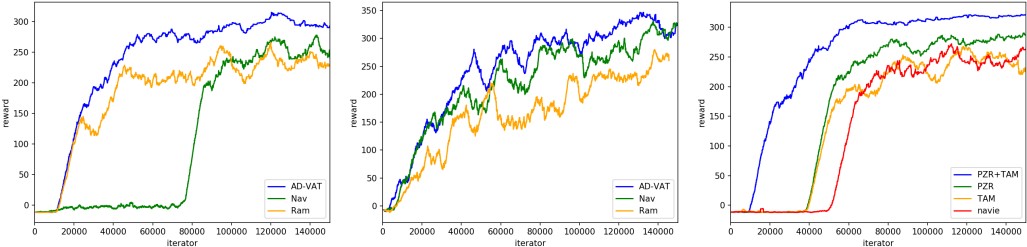

Figure 3: Cumulative reward curves for tracker trained in AD-VAT versus the baseline methods when tested in the validation environment. For both 2D (left) and 3D (middle) settings, AD-VAT improves the sample-efficiency and achieves higher reward. The right figure is the learning curve of the ablation study in 2D environment.

Table 1: Results on the 2D environments.

| Environment | | Accumulated Reward(AR) | | | Episode Length(EL) | | |
|---|---|---|---|---|---|---|---|
| Map | Target | Ram | Nav | AD-VAT | Ram | Nav | AD-VAT |
| *Maze* | Ram | 350±23 | 287±99 | **353±22** | 500±0 | 439±131 | **500±0** |
| | Nav | 243±128 | 257±126 | **264±108** | 412±175 | 409±173 | **431±143** |
| | AD-VAT | 244±129 | 213±125 | **315±25** | 433±145 | 412±154 | **500±0** |
| *Block* | Ram | 352±53 | 303±67 | **362±16** | 491±64 | 462±83 | **500±0** |
| | Nav | 265±134 | 246±144 | **308±60** | 414±178 | 386±194 | **488±69** |
| | AD-VAT | 195±170 | 190±101 | **318±32** | 375±180 | 410±147 | **500±0** |
| Average | | 275±131 | 249±120 | **320±63** | 438±148 | 420±153 | **487±70** |

## 4.4 RESULTS ON THE 2D ENVIRONMENT

We quantitatively evaluate the performance of our approach, comparing to the two baselines. Furthermore, we conduct an ablation study to show the effectiveness of the partial zero-sum reward and tracker-aware model.

**Quantitative Evaluation.** We test the active tracker trained with different target agents in four testing settings, showing the effectiveness of *AD-VAT*. Considering the random seed of the environments, we conduct 100 runs in each and report the mean and standard deviation of AR and EL, shown in Table 1. The max episode length is 500, so the upper bound of EL is 500. Thus, when EL equals to 500, we could infer that the tracker performs perfectly, without losing the target.

We note that, at the beginning of the learning, the adversarial target usually walks randomly around the start point, performing similar policy as *Ram*. Such target is easier to be found and observed, even though the tracker is in exploration. Thus, the tracker could warm up faster. With the growth of the tracker, the target gradually explores other motion patterns, which could further reinforce the tracker. Such a learning process is close to the curriculum learning, but the curriculum is automatically produced by the target via adversarial reinforcement learning. We also report the learning curve as the mean of cumulative rewards in the validation environment, shown as the left sub-figure in Fig. 3. It consistently shows the advantage of the proposed *AD-VAT*.

**Ablation Study.** In Section 3, we introduced two components to implement *AD-VAT*: partial zero-sum reward (PZR) and tracker-aware model (TAM) for target. These two components are important as they influence the natural curriculum for the tracker. Thus, we report an ablation study result to show the effectiveness of these two components, shown in Fig. 3. The naive method is an intuitive idea that target only uses its own observation with auxiliary task, guided by a zero-sum reward. As shown in the right sub-figure of Fig. 3, using each component separately could improve the sample-efficiency, comparing to the naive method. Besides, PZR contributes to the improvement of the tracking performance more significant than TAM. Moreover, when combining PZR and TAM, both sample-efficiency and the tracking performance are significantly boosted, comparing to the other three settings.

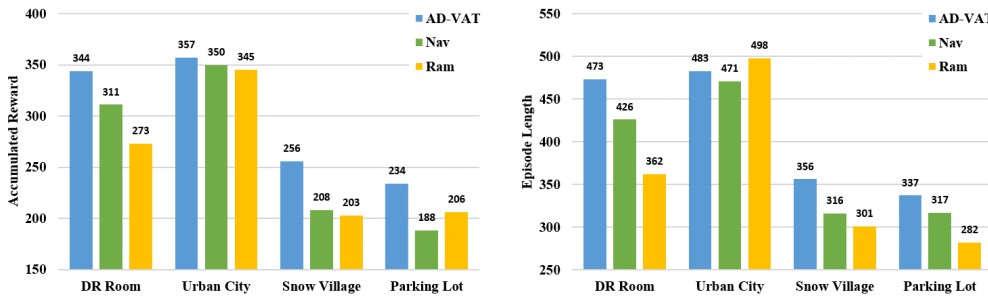

Figure 4: Results on the 3D environment. The higher is the better.

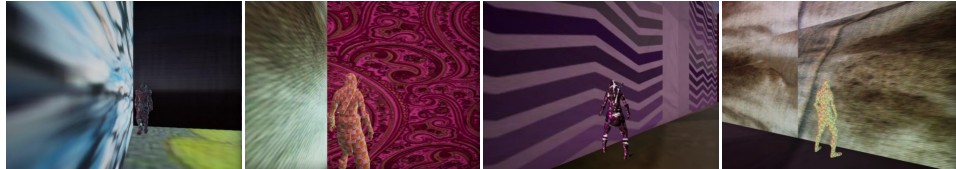

Figure 5: Hard examples produced by the adversarial target.

## 4.5 RESULTS ON THE 3D ENVIRONMENT

In the 3D experiments, we test the generalization of the model in unseen realistic scenarios, showing the transfer potential in real-world scenarios. We train three models with different target models (*Ram, Nav*, and *AD-VAT*) in the DR Room. And then, we directly run the three trackers in the validation and testing environments, 100 episodes for each, and report AR and EL in Fig. 4

The result in DR Room demonstrates again that even though the target is unseen for our *AD-VAT* tracker, it still outperforms the others. Note that the DR Room for validation is of the same settings as the training of *Nav* tracker. The results in the other environments show that the three models are able to transfer to realistic environment. We believe that the domain randomization method and the Conv-LSTM network endow the trackers the ability of transferring. However, the tracker's performance is also highly related to the behavior of the targets during training, especially in complex environments (*Snow Village* and *Parking Lot*). Compared with the two baselines, the adversarial behavior of the target could significantly improve the capability of the tracker in these challenging environments. We infer that the target in *AD-VAT* could explore the environment more acvtively to discover more difficult cases. For example, in DR Room, the target would prefer to move close to the wall that is similar to itself to fool the tracker (see Fig. 5). By competing with the target, the tracker consequently becomes stronger. In Appendix. C, we further evaluate the capability of our tracker in the real-world video clips, qualitatively.

## 5 CONCLUSION AND FUTURE WORK

In this paper, we have proposed an asymmetric dueling mechanism for visual active tracking (AD-VAT). Within AD-VAT, agents of tracker and target are learned in an adversarial manner. With the design of the partial zero-sum reward structure and tracker-aware model, the reinforced active tracker outperforms baseline methods. Experiments including ablation study in both 2D and 3D environments verify the effectiveness of the proposed mechanism.

As future work, we would like to: 1) investigate the theoretical justification of applying modern Multi-Agent RL methods (Lanctot et al., 2017; Srinivasan et al., 2018) to solving Partially Observable Markov Game and finding Nash Equilibrium. 2) further develop the mechanism/model for active tracking in more complex environment (*e.g.*, environments with a number of obstacles and moving distractors); 3) adapt the mechanism to other tasks (*e.g.*, learning to grab a moving object).

## 6 ACKNOWLEDGMENTS

The authors would like to thank Weichao Qiu for his help in extending UnrealCV. Fangwei Zhong, Tingyun Yan and Yizhou Wang were supported in part by the following grants: NSFC-61625201, NSFC-61527804, Tencent AI Lab Rhino-Bird Focused Research Program No.JR201851, Qualcomm University Collaborative Research Program.

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

## A   DETAILS OF THE PARTIAL ZERO-SUM REWARD

|    | $A$ | $\zeta$ | $\xi$ | $\mu$ | $\nu$ | $\rho_2^*$ | $\rho_{max}$ | $\theta^*$ | $\theta_{max}$ |
|----|----|----|----|----|----|------|------|----|------|
| 2D | 1 | 2 | 0 | 1 | 0 | $0grid$ | $6grid$ | \ | $360°$ |
| 3D | 1 | 2 | 2 | 2 | 2 | $2.5m$ | $5.0m$ | 0 | $90°$ |

Table 2: The details of parameters of the rewards in the 2D and 3D experiments.

To help better understand the reward structure given in Eq. (5) and (6), we visualize the sum $r_1 + r_2$ as heatmap in $x - y$ plane. See Fig. 6.

For the 2D experiment, the observations for both the tracker and target are bird-views. We want to penalize that the target gets too far away from the tracker. Therefore, the zero-sum area is a circle (Fig. 6, Left), where the tracker is in the centre. With the increasing of the distance, the penalty term in $r_2$(see Eq. (6)) starts taking effect on the sum. It causes the sum to reduce gradually until the target reaches the dark area, where $r_2 = -A$.

For the 3D experiment, the observations for both the tracker and target are front-views. We want to penalize that the target gets too far away from the tracker or that the target cannot be seen by the tracker. Thus, the zero-sum area is a sector area (Fig. 6, Right), which approximately fits the Field of View (FoV) of the tracker's camera. Note that the FoV in our experiment is 90 degree. Both the relative angle $\theta$ and distance $\rho$ contribute to the penalty term in $r_2$. Thus the sum decreases like a divergence sector.

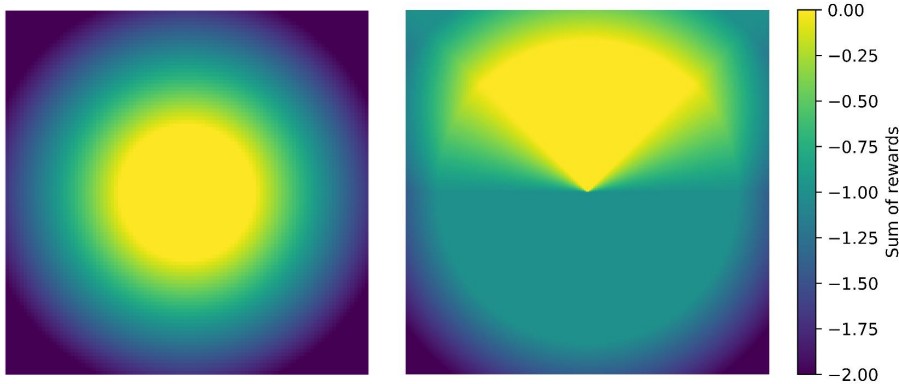

Figure 6:   Visualizing $r_1 + r_2$ as heatmap in $x - y$ plane with the tracker in the image center. Left: The reward adopted in our 2D environment experiment. Right: The reward adopted in our 3D environment experiment. See the Appendix text for more explanations.

## B   VISUALIZING THE TRAINING PROCESS

For a better understanding of the training process, we record trajectories of the target and the tracker during different training stages. Specifically, we have 6 stages, ranging from early training to late training. For each stage we record 100 episodes. In Fig. 7, we plot the target position distribution, instead of the trajectory itself. For ease of visualization, we adopt a relative coordinate system for the target position when drawing the distribution, because the start locations for both the tracker and the target are random upon each episode. In Fig. 7, the distributions are in a *start point*-centric coordinate system, while in Fig. 8 the distributions are in a *tracker*-centric coordinate system.

At early training stages (see the left of Fig. 7), *AD-VAT* and *Ram* generate similar trajectories, which are randomly walking around the start point. In contrast, for *Nav* method the target usually goes along a straight line to the goal position, causing the tracker to get lost quickly at beginning. The random walking trajectories help the tracker observe the target appearing in various positions, and henceforth sample more diverse experiences. As a result, a better exploration is achieved during the early training stage, which is not the case for the *Nav* method.

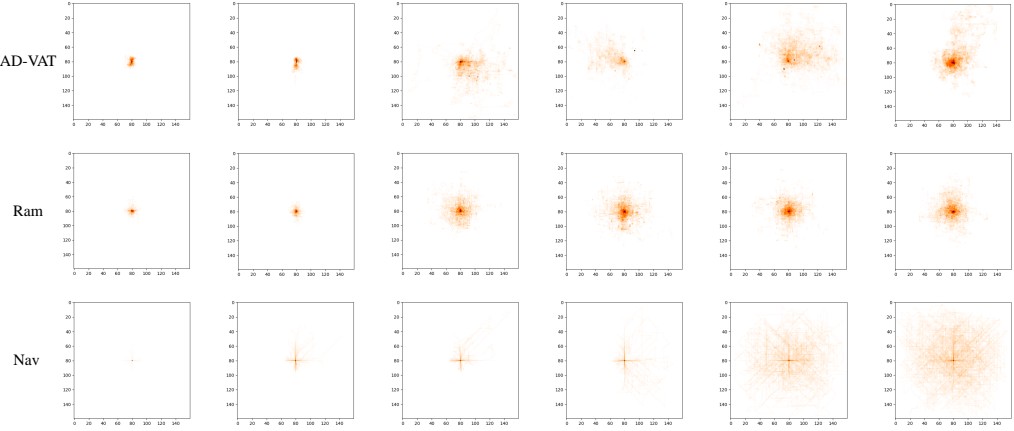

Figure 7: Target position distribution in a *start point*-centric coordinate system. Evolution from early training to late training is arranged from left to right. The darker, the bigger the counts.

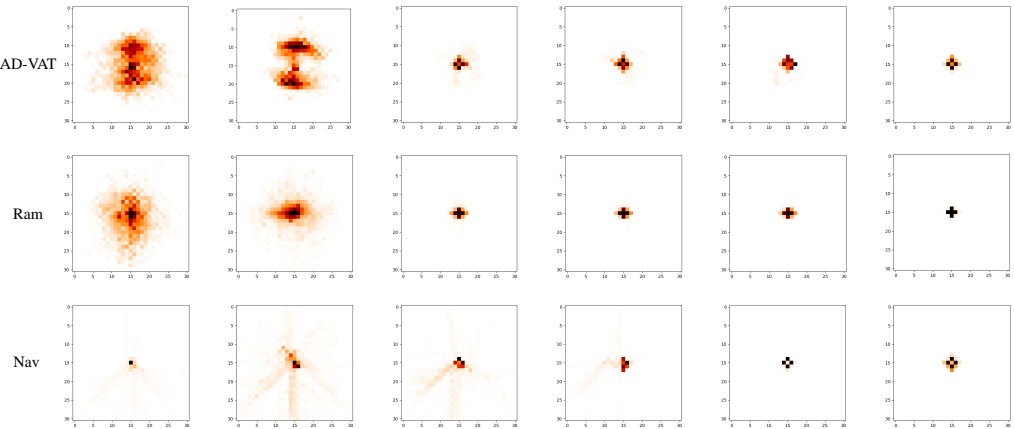

Figure 8: Target position distributions in a *tracker*-centric coordinate system. Evolution from early training to late training is arranged from left to right. The darker, the bigger the counts.

With the evolution of the tracker during training, the target will gradually seek more difficult cases to defeat the tracker. In this regard, the target for both *Ram* and *Nav* usually explores possible directions uniformly. The minor difference is that the *Nav* tends to explore the map globally, while the *Ram* is in local (see the right of Fig. 7). As for our AD-VAT method, however, the reinforced target could adapt to the capability of the tracker, resulting in different direction choosing patterns at different stage. For example, the target tends to move bottom-right at the third stage, but top-left at the fourth stage. See Fig. 7. Besides, it seems that the reinforced target could balance the two exploration modes of *Nav* and *Ram* naturally. Sometimes it explores the map, and sometimes it duels with the tracker locally. By the dueling mechanism, the target could find the weakness of the tracker more often (see the right of Fig. 8), which seems to serve as a kind of importance sampling that enhances the tracker efficiently during training. Such a "weakness-finding" seems absent for the *Nav* and *Ram* algorithms.

## C  TESTING TRACKER ON REAL-WORLD VIDEO CLIPS

To demonstrate the capability of our tracker in real-world scenarios, we conduct a qualitative evaluation as Luo et al. (2018). In this evaluation, we feed the video clips from VOT dataset (Kristan et al., 2016) to the tracker and observe the network output actions. Note that the tracker could not

really control the camera movement, hence we regard the test as "passively". However, the tracker's output action is expected to be sensitive to the position and the scale of the target in the image. For example, when the target appears in the right (left) side, the tracker tends to turn right (left), trying to fictitiously move the camera to "place" the target in the image center. By visualizing whether the output action is consistent with the position and scale of the target at each frame, we are able to demonstrate the potential of transferring the tracking ability to real-world.

We plot three "Action" maps, shown in Fig. 9, Fig. 10, and Fig. 11, respectively. Note that the meaning of each axis is the same as (Luo et al., 2018) except that we normalize the values for better understanding. In details, the horizontal axis indicates the normalized x-axis position of the target in the image, with a positive (negative) value meaning that a target is in the right (left) side. The vertical axis indicates the normalized size of the target, *i.e.*, the area of the ground truth bounding box. We use seven marks to represent the seven actions respectively, as shown in the legend.

More results on other VOT videos are available at: https://youtu.be/jv-5HVg_Sf4.

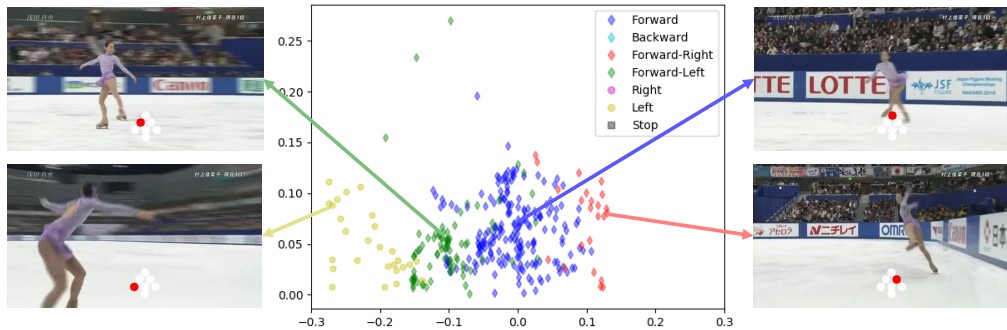

Figure 9: The Action map from AD-VAT tracker feed by the VOT2015-iceskater sequences.

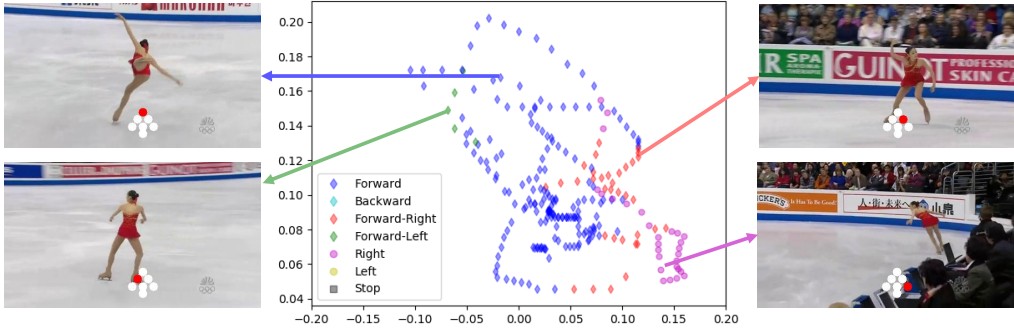

Figure 10: The Action map from AD-VAT tracker feed by the VOT2013-iceskater1 sequences.

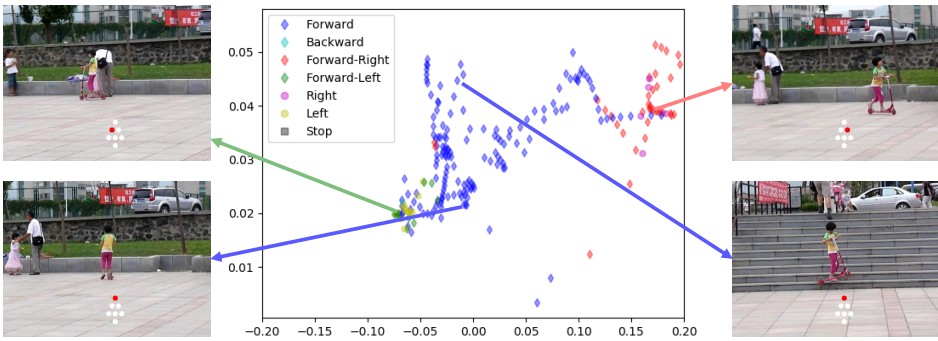

Figure 11: The Action map from AD-VAT tracker feed by the VOT2015-girl sequences.

