# OpenReview forum: "AD-VAT: An Asymmetric Dueling mechanism for learning Visual Active Tracking"
_ICLR.cc/2019/Conference_

### Official Review · AnonReviewer3 · 2018-11-03
**novel reward function in adversarial VAT appliation**

**Rating:** 6
**Confidence:** 4

**Review:**

This is in a visual active tracking application. The paper proposes a novel reward function - "partial zero sum", which only encourages the tracker-target competition when they are close and penalizes whey they are too far.

This is a very interesting problem and I see why their contribution could improve the system performance.

Clarity: the paper is well-written. I also like how the author provides both formulas and a lot of details on implementation of the end-to-end system.

Originality: Most of the components are pretty standard, however I value the part that seems pretty novel to me - which is the "partial zero-sum" idea.

Evaluation: the result obtained from the simulated environment in 2d and 3d are convincing. However, if 1) real-world test and results  2) a stronger baseline can be used, that would be a stronger acceptance.

---

> ### Author Response · Authors · 2018-11-09
> **Reply to AnonReviewer3**
>
> Thanks for appreciating our partial-zero-sum idea. Our primary contribution is the adversary/dueling RL mechanism for training a robust tracker. To stabilize and accelerates the training, we devised the techniques of the partial-zero-sum and the asymmetrical target model. These two techniques are critical for a successful training, and we hope to see their applications to other domains involving adversary/dueling training.
>
> As for the comments on "real-world test and results", we've taken a qualitative testing on some real-world video clips from VOT dataset [Kristan et al. (2016)]. In this evaluation, we feed the video clips to the tracker and observe the network output actions. In general, the results show that the output action is consistent with the position and scale of the target. For example, when the target moves from the image center to the left until disappearing, the tracker outputs actions ``move forward", ``move forward-left", and ``turn left" sequentially. The testing demonstrates the potential of transferring the tracking ability to real-world.
>
> Please see Appendix.C in our updated submission and watch the demo video here: https://youtu.be/jv-5HVg_Sf4

---

### Official Review · AnonReviewer1 · 2018-11-03
**Contrived task**

**Rating:** 4
**Confidence:** 3

**Review:**

This paper presents a simple multi-agent Deep RL task where a moving tracker tries to follow a moving target. The tracker receives, from its own perspective, partially observed visual information o_t^{alpha} about the target (e.g., an image that may show the target) and the target receives both observations from its own perspective o_t^{beta} and a copy of the information from the tracker's perspective. Both agents are standard convnet + LSTM neural architectures trained using A3C and are evaluated in 2D and 3D environments. The reward function is not completely zero-sum, as the tracked agent's reward vanishes when it gets too far from a reference point in the maze.

The work is very incremental over Luo et al (2018) "End-to-end Active Object Tracking and Its Real-world Deployment via Reinforcement Learning", as the only two additions are extra observations o_t^{alpha} for the target, and a reward function that has a fudge factor when the target gets too far away. Citing Sun Tzu's "Art of War" (please use the correct citation format) is not convincing enough for adding the tracker's observations as inputs for the target agent. Should not the asymmetrical relationship work the other way round, with the tracker knowing more about the target?

Experiments are conducted using two baselines for the target agent, one a random walk and another an agent that navigates to a target according to a shortest path planning algorithm. The ablation study shows that the tracker-aware observations and a target's reward structure that penalizes when it gets too far do help the tracker's performance, and that training the target agent helps the tracker agent achieve higher scores. The improvement is however quite small and the task is ad-hoc.

The paper would have benefitted from a proper analysis of the trajectories taken by the adversarial target as opposed to the heuristic ones, and from comparison with non-RL state-of-the-art on tracking tasks. Further multi-agent tasks could also have been considered, such as capture the flag tasks as in "Human-level performance in first-person multiplayer games with population-based deep reinforcement learning".

---

> ### Author Response · Authors · 2018-11-06
> **Thanks for the review.**
>
> Thanks for the review. Our feedback goes below.
>
> Q1: "Contrived task"
> A1: Visual object tracking is widely recognized as an important task in Computer Vision. In this study, we propose a principled approach of how to train a robust tracker.
>
>
> Q2: "The work is very incremental over Luo et al. (2018) "End-to-end Active Object Tracking and Its Real-world Deployment via Reinforcement Learning", as the only two additions are extra observations o_t^{alpha} for the target, and a reward function that has a fudge factor when the target gets too far away"
> A2: Our method is fundamentally different from Luo et al. (2018), as explained below.
>
> Luo et al. (2018) adopted pre-defined target object moving path, coded in hand-tuned scripts. Thus, only the tracker is trainable, and the settings are single-agent RL.
>
> In our method, the target object is also implemented by a neural network, learning how to escape the tracker during training. Both the tracker and the target object are trained jointly in an adversary/dueling way, and the settings are multi-agent RL.
>
> We show the advantage of our method over Luo et al. (2018). Note that the pre-defined target object moving path in Luo et al. (2018) can hurt the generalizability of the tracker. In reality, the target object can move in various hard patterns: Z-turn, U-turn, sudden stop, walk-towards-wall-then-turn, etc., which can pose non-trivial difficulties to the tracker during both training and deployment. Moreover, such moving patterns are difficult to be thoroughly covered and coded by the hand-tuning scripts as in Luo et al. (2018).
>
> The trainable target object in our method, however, can learn the proper moving path in order to escape from the tracker solely by the adversary/dueling training, without hand-tuned path. The smart target object, in turn, induces a tracker that well follows the target no matter how wild the target object moves. Eventually, we obtain a much stronger tracker than that of Luo et al. (2018), achieving the very purpose of our study: to train a robust tracker for VAT task.
>
>
> Q3: "Should not the asymmetrical relationship work the other way round, with the tracker knowing more about the target?"
> A3: We should not do that.
>
> Note that the additional "asymmetrical" information is way of "cheating". As our goal is to train a tracker, we don't need to consider deploying a target object. Therefore, we can simply let the target object cheat during training by feeding to it the tracker's observation/reward/action. Such a "peeking" treatment accelerates the training and ultimately improves the tracker's training quality, as is shown in the submitted paper.
>
> The tracker, however, is unable to "cheat" when deployed (e.g., in a real-world robot). It has to predict the action using its own observations. There is no way for the tracker to acquire the information (observation/reward/action) from a target object.
>
>
> Q4: "The paper would have benefitted from a proper analysis of the trajectories taken by the adversarial target as opposed to the heuristic ones, ..."
> A4: We have added to Appendix some texts for the analysis, see Appendix.B in the updated submission. The target object does show intriguing behaviors when escaping the tracker, see the supplementary videos available at https://www.youtube.com/playlist?list=PL9rZj4Mea7wOZkdajK1TsprRg8iUf51BS
>
>
> Q5: "...and from comparison with non-RL state-of-the-art on tracking tasks."
> A5: Luo et al. (2018) had done the comparisons and shown their method improves over several representative non-RL trackers in the literature.
> Our method outperforms that of Luo et al. (2018).
>
>
> Q6: "Citing Sun Tzu's "Art of War" (please use the correct citation format)..."
> A6: We have fixed this in the updated submission.
>
>
> Q7: "Further multi-agent tasks could also have been considered, such as capture the flag tasks as in "Human-level performance in first-person multiplayer games with population-based deep reinforcement learning""
> A7: The method developed in that paper is for playing the First Person Shooting game, where it has to ensure the fairness among the intra- and inter-team players. In our study, the primary goal is to train a tracker (player 1), permitting us to leverage the asymmetrical mechanism for the target object (player 2). This technique effectively improves the adversary/dueling training and eventually produces a strong tracker.

---

### Official Review · AnonReviewer2 · 2018-11-07
**Incremental contribution and unclear rationales**

**Rating:** 5
**Confidence:** 4

**Review:**

This work aims to address the visual active tracking problem in which the tracker is automatically adjusted to follow the target. A training mechanism in which tracker and the target serve as mutual opponents is derived to learning the active tracker. Experimental evaluation in both 2D and 3D environments is conducted.

I think the contributions of this work is incremental compared with [Luo et al (2018)] in which the major difference is the partial zero sum reward structure is used and the observations and actions information from the tracker are incorporated into the target network, while the network architecture is quite similar to [Luo et al (2018)].
In addition, the explanation about importance of the tracker awareness to the target network seems not sufficient. The ancient Chinese proverb is not a good explanation. It would be better if some theoretical support can be provided for such design.

For active object tracking in real-world/3D environment, designing the reward function only based on the distance between the expected position and the tracked object position can not well reflect the tracker capacity. The scale changes of the target should also be considered when designing the reward function of the tracker. However, the proposed method does not consider the issue, and the evaluation using the reward function based on the position distance may not be sufficient.

---

> ### Author Response · Authors · 2018-11-08
> **Reply to AnonReviewer2**
>
> Thanks for the review. Our feedback goes below.
>
> Q1: "I think the contributions of this work is incremental compared with [Luo et al (2018)] in which the major difference is the partial zero sum reward structure is used and the observations and actions information from the tracker are incorporated into the target network"
> A1:  Our method is fundamentally different from Luo et al. (2018), please see our reply to R#1 (the Q2-A2) for detailed explanations. In short, the major difference is that we employ Multi-Agent RL to train both the tracker and the target object, while Luo et al. (2018) only train the tracker with Single-Agent RL (where they pre-define/hand-tune the moving path for the target object). Our method turns out better in the sense that it produces a stronger tracker via the proposed asymmetrical dueling training.
>
> The Multi-Agent RL training in our VAT task is unstable and slow to converge. To address these issues, we derived the two techniques: the partial zero sum and the asymmetrical target object model.
>
>
> Q2: "In addition, the explanation about importance of the tracker awareness to the target network seems not sufficient. The ancient Chinese proverb is not a good explanation. It would be better if some theoretical support can be provided for such design."
> A2: The tracker awareness mechanism for the target object is "cheating". This way, the target object would appear to be "stronger" than the tracker as it knows what the tracker knows. Such a treatment accelerates the training by inducing a reasonable curriculum to the tracker and finally helps training a much stronger and more generalizable tracker. Note we cannot apply this trick to the tracker as it cannot cheat when deploying. See also our reply to R#1 (Q3-A3).
>
> As for the details of the tracker-aware model, it not only uses the observation and action of the tracker as extra input information but also employs an auxiliary task to predict the tracker's immediate reward. The auxiliary task could help the tracker learn a better representation for the adversarial policy to challenge the tracker.
>
>
> Q3: "For active object tracking in real-world/3D environment, designing the reward function only based on the distance between the expected position and the tracked object position can not well reflect the tracker capacity. The scale changes of the target should also be considered when designing the reward function of the tracker. However, the proposed method does not consider the issue, and the evaluation using the reward function based on the position distance may not be sufficient."
> A3: The scale of a target object showing up in the tracker's image observation will be implied by the distance between tracker and object, which we've considered when designing the reward function.
>
> Consider a simple case of projecting a line in 3D space onto a camera plane. The length (l) of the line on the 2D image plane is derived by an equation as below:
>                                                                                       l = L*f/d,
> where L is the original length in 3D space, f is the distance between the 2D plane and the focal center, and d is the distance between the line and the focal center.
> In the VAT problem，f depends on the intrinsic parameters of the camera model, which is fixed; L depends on the 3D model of the target object, which also could be regarded as constant. Thus, the scale of the object in the 2D image plane is impacted only by d, the distance between the target and the tracker. It is not difficult to derive that, the farther the distance d is, the smaller the target is observed. This suggests that the designed distance-based reward function has well considered the scale of the object.
>
> Note that calculating the scale of the target in an image is of high computational complexity. It requires to extract the object mask and calculate the area of the mask. In contrast, our distance-based reward is computationally cheap, thanks to the simulator's APIs by which we can easily access the tracker's and target's world coordinate in the bird view map.

---

### Author Response · Authors · 2018-11-09
**Change Logs**

We have updated our paper during the rebuttal period, which could be summarized as below:

a) To emphasize our major contribution and clarify the non-trivial different with Luo et al. (2018), we've rewritten Abstract and modified the Introduction.
b) We've modified Section 3.3. The motivation for the tracker-awareness is added. Explanations are given for why we cannot do a target-aware tracker.
c) Supplementary videos are updated in:        https://www.youtube.com/playlist?list=PL9rZj4Mea7wOZkdajK1TsprRg8iUf51BS
  The videos contains:
  1. Training the target and tracker jointly via AD-VAT (2D);
  2. Testing the AD-VAT tracker in four testing environments (2D);
  3. Using the learned target to attack the baseline trackers (2D);
  4. Training the target and tracker via AD-VAT in DR Room (3D);
  5. Testing the tracker in Realistic Environments (3D);
  6. Passively testing tracker on real-world video clips.
d) Appendix.A is modified for better explaining the partial zero-sum reward.
e) Appendix.B is added. It visualizes the training process via drawing the position distribution in different training stages.
f) Appendix.C is added. It provides evaluation results on video clips to demonstrate the potential of transferring the tracking ability to the real world.
g) Table.1 is updated. We add the testing result that the adversarial target is tracked by the three trackers in two different maps, and update the average performance simultaneously. The results demonstrate that the target learned in AD-VAT could effectively challenge the two baseline trackers.

---

### Meta-Review · Area_Chair1 · 2018-12-14
**adversarial learning for active visual tracking with interesting components**

**Confidence:** 5
**Recommendation:** Accept (Poster)

**Metareview:**

The paper presents an adversarial learning framework for active visual tracking, a tracking setup where the tracker has camera control in order to follow a target object. The paper builds upon Luo et al. 2018 and proposes jointly learning  tracker and target policies (as opposed to tracker policy alone). This automatically creates a curriculum of target trajectory difficulty, as opposed to the engineer designing the target trajectories. The paper further proposes a method for preventing the target to fast outperform the tracker and thus cause his policy to plateau. Experiments presented justify the problem formulation and design choices, and outperform Luo et al. . The task considered is  very important, active surveillance with drones is just one sue case.

A downside of the paper is that certain sentences have English mistakes, such as this one:  "The authors learn a policy that maps raw-pixel observation to control signal straightly with a Conv-LSTM network. Not only can it save
the effort in tuning an extra camera controller, but also does it outperform the..." However, overall the manuscript is well written, well structured, and easy to follow. The authors are encouraged to correct any remaining English mistakes in the manuscript.